# Performance and Biomass Characteristics of SBRs Treating High-Salinity Wastewater at Presence of Anionic Surfactants

**DOI:** 10.3390/ijerph17082689

**Published:** 2020-04-14

**Authors:** Huiru Li, Shaohua Wu, Chunping Yang

**Affiliations:** 1College of Environmental Science and Engineering, Hunan University and Key Laboratory of Environmental Biology and Pollution Control (Hunan University), Ministry of Education, Changsha, Hunan 410082, China; lihuiru@hnu.edu.cn (H.L.); wushaohua@hnu.edu.cn (S.W.); 2Guangdong Provincial Key Laboratory of Petrochemical Pollution Processes and Control, School of Environmental Science and Engineering, Guangdong University of Petrochemical Technology, Maoming, Guangdong 525000, China; 3Hunan Provincial Environmental Protection Engineering Center for Organic Pollution Control of Urban Water and Wastewater, Changsha, Hunan 410001, China

**Keywords:** saline wastewater, SBR, activated sludge, extracellular polymeric substance, microbial community

## Abstract

Sodium dodecylbenzene sulfonate (SDBS) and sodium dodecyl sulfate (SDS), as two anionic surfactants, have diffused into environments such as surface water and ground water due to extensive and improper use. The effects on the removal performance and microbial community of sequencing batch reactors (SBRs) need to be investigated in the treatment of saline wastewater containing 20 g/L NaCl. The presence of SDS and SDBS could decrease the removal efficiencies of ammonia nitrogen and total phosphorus, and the effect of SDS was more significant. The effect of surfactants on the removal mainly occurred during the aeration phase. Adding SDS and SDBS can reduce the content of extracellular polymeric substances (EPS). In addition, SDS and SDBS also can reduce the inhibition of high salinity on sludge activity. A total of 16 s of rRNA sequencing analysis showed that the addition of surfactants reduced the diversity of microbial communities; besides, the relative abundance value of the dominant population *Proteobacteria* increased from 91.66% to 97.12% and 93.48% when SDS and SDBS were added into the system, respectively.

## 1. Introduction

Although more than 70% of Earth’s surface is covered by water, freshwater that can be directly used for human activities is less than 1% of global water volume [1]. Due to the increasingly severe situations of freshwater shortage, the direct application of seawater to industrial production and daily life has become a trend. In some industries, such as the printing and dyeing industry, the presence of salt can promote dyeing and improve the quality of textiles. The direct utilization of seawater in the printing and dyeing industry can greatly reduce production costs. The increase of such industries has reduced the loss of fresh water and produced a large amount of saline wastewater. As a result, the discharge of saline wastewater from chemical plants, smelters, and oil production plants has significantly increased in the last decade [2]. If the saline wastewater is discharged directly into the environment without proper treatment, it will detrimentally affect the soil, surface water, and groundwater [3]. Wastewater treatment plants [4] are important systems to prevent the spread of harmful materials to the environment [5,6]. There are many ways to remove pollutants from wastewater [7,8,9]. Physicochemical and biological methods are widely used to treat high-salinity wastewater [10]. The former mainly includes evaporation, electrolysis, coagulation and sedimentation, and membrane separation technology [11]. Unfortunately, the high-cost and potential secondary pollution associated with these methods hampered their applications. Taking environmental protection and economic cost into account, the biological treatment of wastewater has gained popularity owing to the advantages of no secondary pollution [12,13]. On the other hand, the biological treatment of saline wastewater is facing enormous challenges [2]. It was reported that high salinity had an adverse impact on the microorganisms in the wastewater treatment systems [14]. Case in point, when salinity increases, microbial metabolism is inhibited and the removal efficiencies of total phosphorus (TP) and ammonia nitrogen (NH_4_^+^-N) by activated sludge decreases [15,16].

Among the various methods for treating high-salinity wastewater, the biological treatment process has been paid more and more attention due to its low cost and environmental friendliness [17]. To date, sequencing batch reactors (SBRs) have been one of the most widely used biological processes in the wastewater treatment plants. Chen et al. [18] reported that the removal performance of activated sludge in SBRs was significantly inhibited as the salinity was as high as 20 g/L.

Direct seawater utilization wastewater, such as printing and dyeing wastewater, not only contains high salt content, but also some surfactants. Salt can increase the interfacial activity of surfactants in solution [19]. Previous studies have shown that the presence of salt can enhance the adsorption of surfactants, sodium dodecyl sulfate (SDS), and Sodium dodecylbenzene sulfonate (SDBS) at low concentrations [20]. SDS and SDBS are two commonly used anionic surfactants. SDS has excellent foaming properties. SDBS has good emulsifying properties. They have the characteristics of being low cost and playing an important role in industrial production and daily life. They can easily enter the environment through industrial wastewater and domestic sewage [21,22] and may contribute to occurrence and potential environmental risk [23]. SDS and SDBS have an inhibitory effect on the anaerobic ammonia oxidation of microorganisms [24]. It would be positive to indicate the main consequences of the effect of SDBS regarding the increased solubility of proteins and carbohydrates over the activated sludge reaction. When the dosage of SDBS is 0.02 g/g dry sludge, the activity of amylase and proteinase is the best [25]. Moreover, the interaction between salt and surfactant can affect the interfacial tension of the surfactant [26], thus affecting the sludge activity. However, the role of surfactant on the activated sludge system is still unclear, especially under the high salt conditions.

In this paper, the effects of surfactants on the treatment of high-salt wastewater by SBRs were systematically studied, and the nitrogen and phosphorus removal effects of SBRs in the treatment of high-salt wastewater in the presence of surfactants were investigated. The effects of surfactants on the physicochemical properties and microorganisms of activated sludge were analyzed. Through the changes of various parameters, the interaction between salt, surfactant, and microbial flora in activated sludge is analyzed and confirmed. The results show that the presence of surfactant (SDS and SDBS) inhibits the removal efficiencies of nitrogen and phosphorus and promotes the selection of microbial dominant bacteria, which can provide a theoretical basis for the deep understanding of the interactions among high salt, surfactant, and activated sludge.

## 2. Materials and Methods

### 2.1. Water Sample Configuration

The activated sludge used in SBRs in this experiment was taken from Kaifu Sewage Treatment Plant in Changsha City, Hunan province, China. The formula for the synthetic wastewater used in SBRs is as follows: 94.39 mg/L (NH_4_)_2_SO_4_ as a nitrogen source, 460 mg/L Glucose as a carbon source, 17.57 mg/L KH_2_PO_4_ as a phosphorus source, and 0.5 mL/L trace element concentrate. The ingredients in 1 L of concentrate include the following: 1.5 g FeCl_3_·6H_2_O, 0.12 g ZnSO_4_·7H_2_O, 0.15 g H_3_BO_3_, 0.06 g Na_2_MoO_4_·2H_2_O, 0.15 g CoCl_2_·6H_2_O, 0.18 g KI, 0.03 g CuSO_4_·5H_2_O, 10 g EDTA, and 0.12 g MnCl_2_·4H_2_O [18,27]. The high salt wastewater is obtained by diluting 20 g/L NaCl with tap water. The concentration of surfactants was 0.08 g/L. The compound comprised to two kinds of synthetic wastewater, one with SDS and another with SDBS, as well as the control with the same formula but without any surfactant.

### 2.2. Operation of Reactors

This study was carried out in three SBRs, and each reactor had an effective volume of 3 L. Three SBRs were all operated with the influent salinity of 20 g/L. Filler studies were performed in three stable SBRs. SBR1 was the control without the addition of surfactants, while SBR2 and SBR3 were dosed with 0.08 g/L SDS and SDBS, respectively. The initial concentrations of mixed liquor suspended solids in SBR 1, SBR 2, and SBR 3 were 4500 mg/L.

There were three operation cycles with the SBRs per day. The flow of each cycle was as follows: 0.05 h for feeding, 3.5 h of aeration reaction after 1 h of standing, 2 h for stirring, 0.5 h for settling, 0.05 h for decanting, and 0.9 h of idle period. Although the composition of the influent streams was different, the operating conditions of each reactor were the same. The duration of each step in the reaction was regulated by the time controllers. Two peristaltic pumps were used to feed and draw the influent and effluent of SBRs. The temperature of the reactors was controlled by temperature conditioners and the temperature in the reactors were maintained at 25 ℃. During the aeration phase, aeration is achieved by an electromagnetic air compressor. During the anoxic phase, the aeration was stopped, a magnetic stirrer was used to stir, and the system was maintained in an anoxic environment.

Analytical methods and 16S rRNA sequencing analysis are available in the Appendix A. Among them, the 16S rRNA gene amplicon products were sequenced using the HiSeq2500 platform at Illumina. Moreover, the analysis methods of Alpha diversity and beta diversity of microbial community are also included.

## 3. Results and Discussion

Appendix A shows the effect of surfactant addition on the removal rate of NH_4_^+^-N and TP in the SBRs system. Among the three stable SBRs, the surfactants SDS and SDBS were added to SBR2 and SBR3, respectively. After about 35 days, the removal rates of NH_4_^+^-N and TP in the three SBRs systems reached stability again. Maintain a steady state for a period of time and select a typical operating cycle for more in-depth inspection research.

### 3.1. Removal Performance Analysis

The N and P removal performance of SBRs is presented in Figure 1a,b. Overall, comparing with the control, the addition of surfactants affected mainly the removal rates of NH_4_^+^-N and TP [28]. The removal of NH_4_^+^-N was inhibited in the presence of SDS and SDBS, and the related removal efficiency decreased from 63.41% to 42.57% and 59.81%, respectively. Moreover, the concentration profiles of NO_2_^−^-N and NO_3_^−^-N within one operational cycle were also monitored. Figure 1c,d shows the variation of NO_2_^−^-N, NO_3_^−^-N during an operation cycle, respectively. Under high salt conditions, the presence or absence of surfactants has basically the same effect on the content of NO_2_^−^-N in SBR. Nevertheless, in the system with SDS and SDBS, the nitrate nitrogen content increased significantly during the reaction and nitrification and denitrification enhancement [29]. It was indicated that nitrification and denitrification have a great influence on the denitrification efficiency of the system [30,31,32]. The comparison of Figure 1a,d showed that that the concentration of NO_3_^−^-N is the highest, while the removal rate of NH_4_^+^-N is the lowest, in the presence of SDS. It was found that the activated sludge in SBRs was affected not only by surfactants [23,33], but also by high salinity [20,26,34]. High salinity has a strong inhibition effect on sludge activity [18], but the interaction between surfactant and salt weakens the inhibition of salt on the activated sludge. Figure 1b has shown that the effect of the presence of two surfactants on the TP removal rate in the system. Considering the kinetic factors, by fitting equations to the experimental data, among the three sets of data, the TP removal rate changes with the time (t) during the operation cycle and satisfies the following formulas:r_(control)_ = − 40.03715 + 48.19004 × t − 9.11171 × t^2^ + 0.76962 × t^3^ − 0.02466 × t^4^(1)
r_(SDS)_ = − 48.49176 + 67.42909 × t − 23.16313 × t^2^ + 3.65602 × t^3^ − 0.20901 × t^4^(2)
r_(SBDS)_ = − 22.58669 + 24.41505 × t − 5.28091 × t^2^ + 0.94579 × t^3^ − 0.06628 × t^4^(3)
where r_(control)_, r_(SDS)_ and r_(SBDS)_ are the removal rate of TP in the control system, the SDS-containing system, and the SDBS-containing system, respectively. The quality of the fit polynomial model was expressed by the coefficient of determination R^2^. It can be seen from Figure 1b that the removal rate of TP hardly changed during the standing period, and, when aeration started, the three lines have a good fitting effect. The addition of SDS reduced the removal rate of TP from 55.64% to 41.34%, while the addition of SDBS only inhibited the removal of TP in the initial stage of the reaction; after a short lag phase, the removal rate of TP rose to 55.06%, the inhibition effect weakened, and, ultimately, TP reverted to the same level as the control. This is probably because the addition of anionic surfactants inhibited the reaction when phosphorus accumulating organisms use oxygen as electron acceptor to remove TP. Additionally, due to the difference in molecular size and stability between SDS and SDBS, the inhibitory effects on SDS and SDBS are different [35]. When the removal efficiencies of NH_4_^+^-N and TP were stable, the removal performance of nitrogen and phosphorus in the system containing SDS and SDBS were inhibited. From previous studies, the kinetic studies on the removal rates of ammonia nitrogen and TP in the SBRs system are less than nitrate nitrogen and nitrite nitrogen. However, from the comparison of the removal rate, since this experiment is to add surfactant under high salt conditions where the activated sludge can maintain activity, this result is quite impressive [1,18,31]. The reduction in nitrogen and phosphorus removal during the reaction may be due to the effects of salts and surfactants. The pressure of the cells forces the functional proteins on the cell membrane to increase, and the secretion of glycogen and nucleic acids in the cells is reduced [36]. Glycogen contributes to the storage of nitrogen, and its reduction causes the inhibition of nitrification and denitrification, and nitrogen removal efficiency decreases [37,38]. Similarly, the reduction in nucleic acid reduces the removal rate of phosphorus [36]. Meanwhile, the increase in Chemical Oxygen Demand (COD) in the three systems (control, SDS-containing system and SDBS-containing system) were 82.32 mg/L, 156.45 mg/L, and 195.49 mg/L, respectively. This may be due to the stress of high salinity and surfactants, resulting in the rupture of some cell membranes during the reaction process. The decrease in SS (suspended solids) content in the system during continuous reaction may also due to this. Theoretically, N and P and COD will be converted into nutrients during the operation of the SBR system [39], promoting the growth of microorganisms and the production of organic matter in the activated sludge, and increasing SS [40,41]. However, in the actual reaction, due to the high salt and surfactant stress, some microbial cells ruptured, the SS in the system decreased, and the COD in the effluent increased.

### 3.2. Activated Sludge Performance Analysis

The influence of surfactants on the performance of activated sludge is generally manifested in the sedimentation of activated sludge, the variation of extracellular polymer substance (EPS), and microbial enzyme activities. These three characteristics of activated sludge were closely related to each other [42]. For instance, the composition and proportion of EPS have a great influence on microbial enzyme activity and sludge sedimentation. EPS is composed of proteins (PN), polysaccharides (PS), and nucleic acids. Among them, 70%–80% of EPSs are proteins and polysaccharides [43,44]. In addition, EPS can enrich nutrients in the environment and has an important influence on flocculation structure and settling performance of activated sludge [45].

In addition, dehydrogenase is an oxidoreductase that can catalyze substances. It can reflect not only the degradation ability of microorganisms but also the operation effect of the reactor in the process of organic matter removal. Variations of the dehydrogenase activity (DHA) under surfactant stress are shown in Figure 2a. The content of DHA in the system with SDBS is 55.65 mg TF/g TSS/h, similar to the 50.57 mg TF/g TSS/h in the control system. The content of DHA in the system with SDS is 226.11 mg TF/g TSS/h, obviously higher than that of the former two, which indicated that the addition of SDBS had little effect on the activity of microbial enzymes. Nevertheless, the activity of microbial enzymes in the SDS-containing system was somehow higher than the control and SDBS-containing system. This is mainly due to the interaction between SDS and salt ions, which reduced the inhibition of high salt on the activity of the microbial enzymes [46]. The variations of EPS contents under surfactant stress were shown in Figure 2b, and the addition of SDS and SDBS could reduce the EPS content of the system from 37.25 to 25.90 and 24.53 mg/g SS, respectively. The content of PN and the loosely bound extracellular polymeric substances (LB-EPS) also decreased, respectively. The content of the tightly bound extracellular polymeric substances (TB-EPS) in the system containing SDBS was almost unchanged, while the content of TB-EPS in the system containing SDS decreased significantly [42].This suggested that the increase in LB-EPS is not conducive to the sedimentation of activated sludge [47]. However, as shown in Figure 2c, the proportion of LB-EPS/EPS in the system containing SDS increased, and the sedimentation of activated sludge increased. The SV30 in the figure refers to the volume percentage of the sludge after the mixed liquid is left in the graduated cylinder for 30 min. A small SV30 value indicates good sludge sedimentation. In the SDBS-containing system, the percentage of LB-EPS/EPS decreased and the sedimentation of activated sludge weakened. There may be two reasons for this result: (1) PS/EPS enhances the ability of PS to protect cells and enhances the sedimentation of activated sludge [42,48]; (2) the interaction between SDS and salt reduces the stress of salt on the system, thus enhancing the sedimentation of activated sludge [46]. SDBS could not reduce the stress of salt on the system, and thereby the activity of enzymes was inhibited. Besides, the interaction between SDBS and salts could not enhance the sedimentation of activated sludge.

### 3.3. Microbial Community Structure Analysis

The alpha diversity index can characterize the complexity of sample species diversity, including population number and the Chao, Shannon, Simpson, and coverage index [18]. The alpha index of three activated sludge samples with the addition of surfactants are presented in Table 1, and the high Good’s Coverage indexes of all three samples are 1, which indicated that the sequencing depth of all samples are sufficient [49]. The Chao 1 in the SDS/SDBS-containing systems was lower than that in the control without surfactants, which suggested that the abundance of microbial community decreased due to the addition of surfactants. As the Shannon index is non-linear, MacArthur et al. correctly identified diversity with exp (H_Shannon), and the effective number of species can more intuitively compare the richness of different biological flora. As shown in Table 1, the sample with SDBS shows a higher richness and evenness than the control does, while SDS does the opposite. Shannon and Simpson can represent the diversity of microbial communities, the variation of their indices showed that the addition of surfactants SDS and SDBS decreased the diversity of microbial communities in the system [49]. Moreover, the addition of SDS and SDBS reduced the observed species from 250 to 139 and 195, respectively. In general, the addition of surfactant reduced the richness of the biological community, as well as the diversity of the biological community.

The Wayne diagram of the microbial community could clearly reflect the effect of two surfactants on SBRs under high salt conditions. Appendix A has shown that the Shared species among the three samples was 103. The Shared species between every two samples, including the control and SDS, the control and SDBS, and SDS and SDBS, were 120, 168, and 112, respectively. This means that under the condition of high salinity, the effects of anionic surfactants on the microbial community diversity of SBRs system were different, and the inhibition of SDBS on the system was less than SDS. This result is in agreement with the fact that the interaction between SDBS and salt cannot reduce the effects of salt stress.

The operational taxonomic unit (OTU) means that the tags clustered at 97% similarity by UPARSE (version 7.1) [36]. Figure 3a shows the distribution of OTU numbers of the three samples at the seven taxonomic levels of *Kingdom, Phylum, Class, Order, Family, Genus,* and *Species*. The OTU numbers at the levels of *phylum, genera,* and *species* of dimension were obviously influenced by surfactants. Figure 3b–d shows the microbial population abundance map of activated sludge samples at the *phylum,*
*genus,* and *species* level under the effect of surfactant. The addition of SDS and SDBS reduced the number of species classified by the system but increased the number of genera. At the phylum level, the dominant floras of all samples were *Proteobacteria*, *Firmicutes*, *Bacteroides*, *Actinomycetes,* and *Saccharibacteria*. *Proteobacteria* was the most abundant one in the three samples. The relative abundance value of *Proteobacteria* increased from 91.66% to 97.12% and 93.48% with the addition of SDS and SDBS, respectively. *Proteobacteria* has strong adaptability under the influence of high salinity and surfactant, and it was reported that the most denitrifying bacteria belonged to *Proteobacteria* [50,51,52]. Figure 3d shows that *Kluyvera* was one of the *Proteobacteria* that played a major role in this study. The lower the relative abundance of *Kluyvera*, the lower the nitrogen removal efficiency [53,54]. However, this phenomenon was not observed in this study. This was due to the fact that the removal efficiency of NH_4_^+^-N is not only affected by effective dominant bacteria but also by the richness and diversity of the microbial community, and the total amount of effective bacteria is the key to nitrogen removal. On the generic level, the dominant genera in the control and SDBS-containing system were *Clovera*, *Candida rubra,* and *Thiothrix*, while the dominant genera of the SDS-containing system were *Clovera* and *Candida rubra*. *Thiofacillus* was beneficial to sulfidation and desulfurization and reduced the impact of SDBS on various indicators of activated sludge in the SBR system. The addition of surfactant reduced the microbial diversity under high salt conditions, but increased the relative abundance of dominant populations. This is one of the reasons why the addition of SDS increased the activity of activated sludge enzymes in the system and increased the content of nitrous nitrogen in the reaction process, but the ability to remove nitrogen and phosphorus decreased.

Figure 4 shows the results of unweighted pair-group method with arithmetic means (UPGMA) analysis on the gate level. The results of UniFrac distance analysis analyzed the diversity of microbial communities by comparing the distance between each two samples and showed that SDS and SDBS have some effects on microbial species. UPGMA analysis based on unweighted UniFrac distance showed that sample SDBS is similar to sample control, which indicates that the difference in microbial population diversity between the SDS-containing system and the control group was greater than that between the SDBS-containing system and the control system. UPGMA analysis based on weighted UniFrac distance showed that the addition of SDS and SDBS not only have effects on microbial species diversity, but also affect the richness of microbial communities. The comparison of relative abundance in phylum level also confirms this conclusion. Therefore, the difference of activated sludge under the influence of surfactant was subjected to the changes in microbial species diversity and microbial community richness. The combination of the inhibitory effect of salt on the sludge activity, the interaction between surfactants and salt, and the effect of surfactants on the microbial community in activated sludge has led to a reduction in the nitrogen and phosphorus removal performance of activated sludge.

## 4. Conclusions

Under high salt conditions, the addition of SDS and SDBS in SBRs can inhibit the removal of nitrogen and phosphorus. The inhibition of SDS is more obvious. Sludge activity was almost affected in the SDBS-containing system, while it increased in the SDS-containing system. The interaction between SDS and salt can reduce salt stress on sludge activity. Besides, the addition of SDS and SDBS reduced the content of the EPS of the system. The content of TB-EPS in the system containing SDBS was almost unchanged, while the content of TB-EPS in the system containing SDS decreased significantly. The 16S rRNA sequencing analysis showed that *Proteobacteria* was the dominant population in the presence of high salt and surfactant. Moreover, the removal performance of SBRs is affected not only by dominant population but also by microbial community richness and community diversity. Therefore, the reduction in the nitrogen and phosphorus removal performance of activated sludge is the result of several factors acting together.

## Figures and Tables

**Figure 1 ijerph-17-02689-f001:**
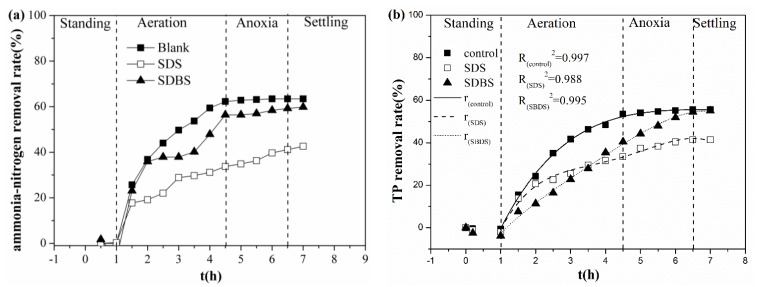
Removal rates of pollutants during an operational cycle: (**a**) NH_4_^+^-N; (**b**) TP; (**c**) NO_2_^-^-N; and (**d**) NO_3_^-^-N.

**Figure 2 ijerph-17-02689-f002:**
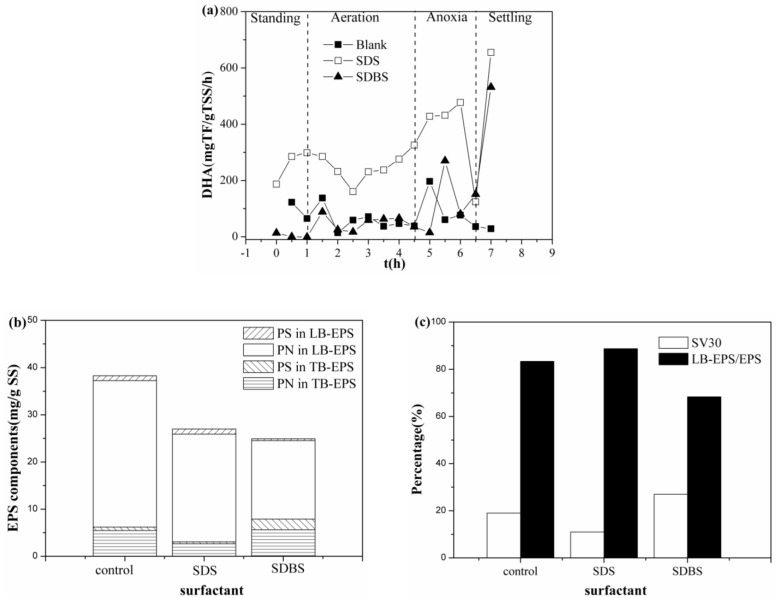
Variation of activated sludge characteristics under different surfactants: (**a**) dehydrogenase activity (DHA); (**b**) extracellular polymer substance (EPS); and (**c**) Sedimentation performance.

**Figure 3 ijerph-17-02689-f003:**
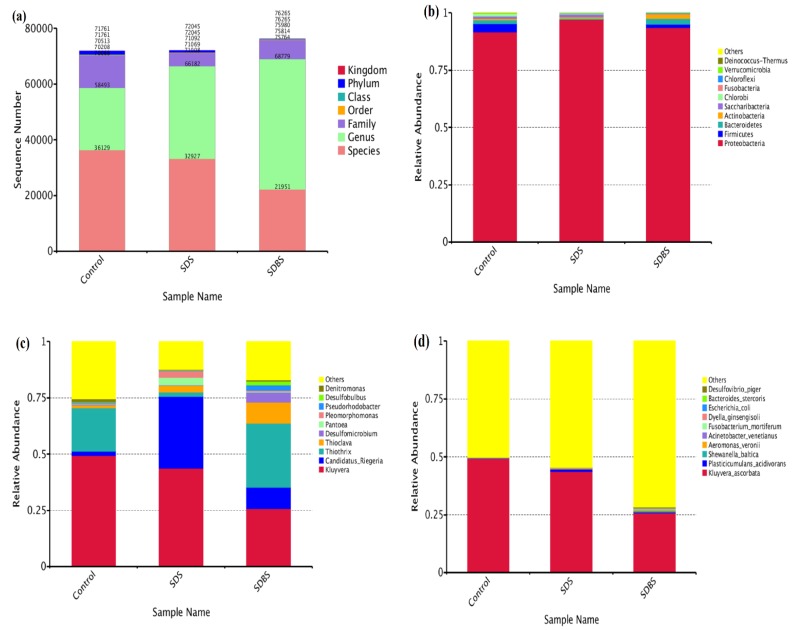
Operational taxonomic unit (OTU) numbers of the 3 samples: (**a**) at the seven taxonomic levels; (**b**) at the phylum level; (**c**) at the genus level; and (**d**) at the species level.

**Figure 4 ijerph-17-02689-f004:**
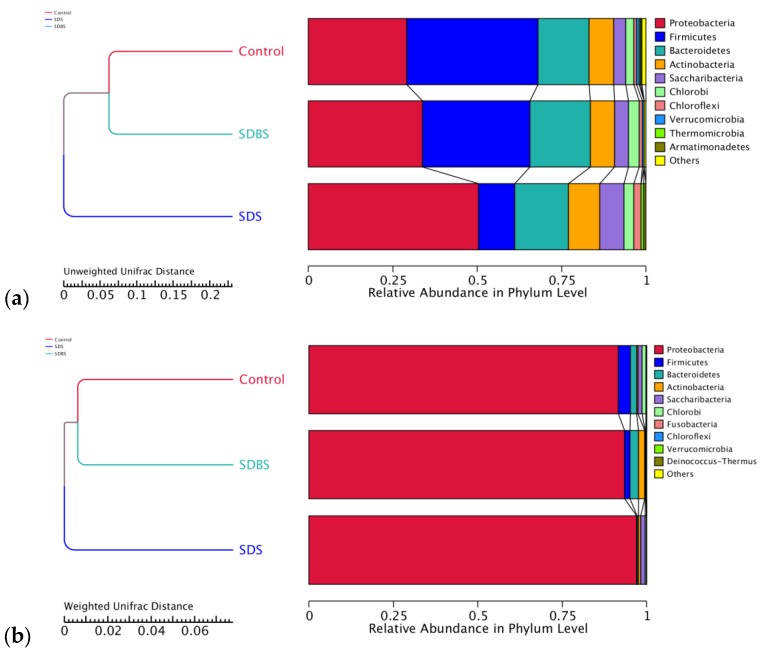
UPGMA analysis based on different algorithms at the phylum level of 3 activated sludge samples: (**a**) the unweighted unifrac distance; (**b**) the weighted unifrac distance.

**Table 1 ijerph-17-02689-t001:** Alpha index of three activated sludge samples with the addition of surfactants.

Sample Name	Observed Species	Shannon	Simpson	Chao1	ACE	Goods’ Coverage
Control	250	2.882	0.700	256.000	256.270	1
SDS	139	2.652	0.702	145.067	147.287	1
SDBS	195	3.450	0.828	206.875	205.442	1

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
