# Peer review of "Performance and Biomass Characteristics of SBRs Treating High-Salinity Wastewater at Presence of Anionic Surfactants"

_ijerph, 2020, doi:10.3390/ijerph17082689_

Round 1

Reviewer 1 Report

Revisionfor ID. ijerph-747789-peer-review-v1.

General comment: I`ve found the manuscript quite interesting and well-structured and presented, but I feel data could be more developed and discussion enlarged regarding the relationship between chemical and biological findings and supported in previous works.

Abstract

Line 18-19: Maybe I would add “in the environment” to the sentence.

Line 26-29: Consider to split in two sentences or reorder the sentences because it remains confused as it is formulated now. (e.g.”…showed that the addition of…reduce……..and increase….…”)

Introduction

Line 56-57: The information given in this sentence is the same as in lines 50-51.

Line 69-70: It would be positive to indicate the main consequences of the effect of SDBS regarding the increased solubility of proteins and carbohydrates over the activated sludge reaction.

Line 70-71: Is that dose considered low or high? Is this dose founded frequently in the polluted environment? Is this value a limit dose?

Materials and Methods

Line 89-90: It should be clear that it is respectively. I mean to let clear-cut enough that there are two kinds of synthetic wastewater, one with SDS and another with SDBS, as well as the blank with the same formula but without any surfactant.

Line 91: Numeration of this subsection should be 2.2 instead of 2.1.

Line 92-94: I see that the information asked in my first comment of the section is here but, at least, I think is necessary to indicate in the previous subsection that the composition explained is for the 3 SBRs with the exceptions of the surfactants. Nevertheless, I would split this sentence (92-94) in two eliminated the first “and”.

Line 98: What do you mean by the “influent streams were different”? Why? Where is that reflected?

In general: I miss information regarding the data acquisition which is used to build the subsequent results and plots presented in the next section. Even when it is explained in the supplementary material some minor information should be reflected in the main text. For example, regarding sequencing data at least mention that Illumina was used; and minor references to data acquisition of rate of pollutants.

Results and Discussion

Line 107: The number of the section is 3 not 2 again. Also review the subsection numbers since there are some incorrect.

Line 114-115: The concentration of NO2 in the blank, which I supposed is the control without surfactants, is equally null than with both surfactants so you cannot assert the effect is due to the presence of surfactants.

Line 117-119: Needs a little more discussion. Explain in which way the subsequent steps of nitrification and denitrification enhance the global denitrification of the system and link it with the lack of negative effect of the surfactants in this case.

Line 123: What Fig1b is showing is already said some previous lines. Try to give the information of the sentence in another way.

Line 124: Take care when use the term “significantly” because it could lead to confusion  making wonder if there is any statistic treatment of the data that indicates statistic significance. Maybe change it in some sentences for noticeable, remarkable or others similar.

Line 132: Data not shown?

Line 134: Please, link the presence of surfactants and salt to the “pressure on cells” with a short explanation.

Line 138: Indicates what COD (Chemical Oxygen Demand) is (do it for any acronym the first time it appears). Increase regarding what? If it is regarding the control the sentence as it is formulated is confused. If it is regarding to other data please explain it.

Line 141: Explain SS content.

Line 153: Fig 2. Explain the acronym DHA also here (I see it is explained later) since it is the first time it appears in the manuscript. The same for EPS. Also explain what SV30 is and indicated it at least once in the main text when talk about the increase/decrease of the activated sludge.

Line 156: Figure 3, in the caption: seven taxonomic levels. Also, the captions at the right of images are too small and some words are cut.

Figures in general: I would place each figure nearby (or after) the corresponding text explanation and reference to each one in the corresponding subsection or paragraph, but not everyone together in the middle of results and discussion section as they are now.

Line 161: Incorrect numeration.

Line 164: What does “these” refer to? Maybe split 162-164 sentence in two or indicate “these three characteristics” to specify and clarify.

Line 176-177: There is a lack of semicolon after “nevertheless”. I find this information repetitive with the previous sentence; maybe reorder the sentences.

Lines 173 and 179: (as say it before) Figures 2 would be much more useful after line 190 or at the beginning of the section but not 3 pages before.

Line 181: Indicate what LB-EPS is.

Line 181-182: I can´t see a clear connection between this increase and the data on the previous sentence regarding decrease; neither with the information afterwards.

Line 192-193: The sentence needs a reference.

Line 195: Good’s Coverage indexes as adapted from Singleton DR, Furlong MA, Rathbun SL, Whitman WB. Quantitative comparisons of 16S rRNA gene sequence libraries from environmental samples. Appl Environ Microbiol. 2001;67(9):4374–4376. doi:10.1128/aem.67.9.4374-4376.2001 when if C is close to 0 there is a lack of diversity detection and close to 1 indicates a good representative sample. Maybe a reference to this paper and a short explanation of what C=1 means will be helpful. Do all of your reads of OTUs appear at least twice in the samples?

Line 195-200: In general, more information and discussion about what the indexes tell is needed. Also, take care with Shannon index since is non-linear (see https://www.sciencedirect.com/science/article/abs/pii/S1470160X16301479 )maybe try to use Effective number of species instead of Shannon (http://www.loujost.com/Statistics%20and%20Physics/Diversity%20and%20Similarity/EffectiveNumberOfSpecies.htm). As presented now the sample with SDBS shows a higher richness and evenness that control does.

Line 209: This results means or This means, disclosed….or another formula to not use “shown” every time.

Line 211: Inhibition of what? Inhibition of diversity? Of number of species? Inhibition of the effect of salt due to the surfactant? Is that in agreement with the potential effect of SDSB on salt exposed in lines 189-190. It would be a good point discuss it.

Line 212: Uncompleted sentence??

Line 213: Add taxonomic to “the seven levels”.

Line 214: dimensionae?

Line 216: respectively?? Please, indicate it. In figure 3c I can see how the number of genus increase in the samples with SDS and SDBS but figure 3d does not show a decrease of number of species regarding control sample.

Line 223: I can see de accuracy of these two references in the sentence unless you enlarge the discussion and connect them

Line 224: This statement needs a reference.

Line 226: Why?

Line 230-234: So, you are saying that the species dominant that increase due to the SDS presence are not involved in the N and P removal? At which taxonomic level do you refer here when talk about diversity?

Line 236: Remove “were”.

Line 235-237: Suggestion to make one from two sentences: “The results of UniFrac distance analysis were analyzed the diversity of microbial communities by comparing the distance between each two samples and showed taht SDS and SDBS have some effects on microbial species.” And then explain which ones these effects are.

Line 237-239: The fact that samples SDBS and control are quite similar regarding unweighted UniFrac distance does not imply (or indicate) that the addition of SDS influences the diversity of microbial population in the system. It should depend on the data regarding comparison between SDS sample and control. Please, reformulate it.

Line 239-242: Good point, but it needs a little more previous discussion to support it. I found a bit of lack in deepness in the general discussion of data obtained and relation between them.

Line 242-244: It would be good to indicate here which are these factors, to complete the sentence, or maybe to move it to Conclusions section.

Conclusions

Line 249: “In addition, addition…”. Try to use Besides or something similar.

Line 250: This is the first time TB-EPS appears in the main text and you should explain/indicate what TB means; and do it the same for LB-EPS (tightly and loosely bound) (it is not enough to let them indicated in the supplementary material which is another document). In general (not specifically here that is conclusion section) I feel there should be more discussion, with references supporting your findings, e.g. the relation between EPS increase/decrease with the addition of SDS because the salinity is affected (your reference number 42 could be helping) (It is one of other examples through the text).

Line 252: Have you found halophiles or differences in the abundance of them between samples?

Line253-255: The sentence is almost the same that the end of previous section. The statement is accurate here but it should be deeper discussed in the previous section.

Supplementary material

“The detailed protocols of these two processes were described according to Chen et al. (2018)” -- Described or applied??

“against the GreenGene database with aconfidence threshold of 80%-- ..with a confidence..

Reviewer 2 Report

The manuscript with the title “Performance and biomass characteristics of sequencing batch reactors treating high-salinity wastewater at presence of anionic surfactants” is an interesting work but I have some comments:

Abstract

There is a lot of percentage data and non-significant find, and let to think the work does not have any new contribution to the scientific and technological community.

Introduction

What type of theoretical basis for the further study of the interaction between high salt, surfactant and activated sludge, you are going to find? I think in Introduction must to talk about this theoretical basis.

Materials and methods

I think a scheme of the different SBR must be presented with dimensions and particularisms

It missing the analytical methods to measure the concentrations of chemical species during the tests.

Results and discussion

The discussion about the percentages of removal and efficiency is very weak. I think if you present experimental data as a function of time you must do a kinetic analysis in order to compare with other works.

In general, all the discussion is only a big observation, none explication of phenomena and none result was evaluated with fundamental knowledge.

Round 2

Reviewer 2 Report

The manuscript with the title “Performance and biomass characteristics of sequencing batch reactors treating high-salinity wastewater at presence of anionic surfactants” is an interesting work but I have some comments:

Abstract

There is a lot of percentage data and non-significant find, and let to think the work does not have any new contribution to the scientific and technological community.

Introduction

What type of theoretical basis for the further study of the interaction between high salt, surfactant and activated sludge, you are going to find? I think in Introduction must to talk about this theoretical basis.

Materials and methods

I think a scheme of the different SBR must be presented with dimensions and particularisms

It missing the analytical methods to measure the concentrations of chemical species during the tests.

Results and discussion

The discussion about the percentages of removal and efficiency is very weak. I think if you present experimental data as a function of time you must do a kinetic analysis in order to compare with other works.

In general, all the discussion is only a big observation, none explication of phenomena and none result was evaluated with fundamental knowledge
